# Current Methods for Reliable Identification of Species in the *Acinetobacter calcoaceticus*–*Acinetobacter baumannii* Complex

**DOI:** 10.3390/microorganisms13081819

**Published:** 2025-08-04

**Authors:** Teodora Vasileva Marinova-Bulgaranova, Hristina Yotova Hitkova, Nikolay Kirilov Balgaranov

**Affiliations:** 1Department of Microbiology and Virology, Medical University of Pleven, 5800 Pleven, Bulgaria; teodora.marinova@mu-pleven.bg; 2Department of Pediatric Diseases, Medical University of Pleven, 5800 Pleven, Bulgaria

**Keywords:** *Acinetobacter baumannii*, identification, MALDI-TOF MS, molecular methods

## Abstract

*Acinetobacter baumannii* is one of the most challenging nosocomial pathogens associated with a variety of hospital infections, such as ventilator-associated pneumonia, wound and urinary tract infections, meningitis, and sepsis, primarily in patients treated in critical care settings. Its classification as a high-priority pathogen is due to the emergence of multidrug-resistant strains in healthcare environments and its tendency to spread clonally. *A. baumannii* belongs to the *Acinetobacter calcoaceticus*–*Acinetobacter baumannii* (Acb) complex, a group of genotypically and phenotypically similar species. Differentiating between the species is important because of their distinct clinical significance. However, conventional phenotypic methods, both manual and automated, often fail to provide accurate species-level identification. This review aims to summarize current phenotypic and genotypic methods for the identification of species within the Acb complex, evaluating their strengths and limitations to offer guidance for their appropriate application in diagnostic settings and epidemiological investigations.

## 1. Introduction

The most clinically significant representative of the genus *Acinetobacter* is *Acinetobacter baumannii*. It is a major cause of healthcare-associated infections, including ventilator-associated pneumonia, bloodstream infections, meningitis, wound infections, and urinary tract infections, particularly among immunocompromised patients. The World Health Organization (WHO) has designated *Acinetobacter baumannii* as a critical priority pathogen due to its extensive resistance to last-line antibiotics and its capacity for clonal dissemination [1].

*A. baumannii* is a member of the *Acinetobacter calcoaceticus*–*Acinetobacter baumannii* complex (Acb complex), a group of phenotypically and genetically closely related species. Epidemiological studies underscore the clinical relevance of other complex members, notably *A. pittii* and *A. nosocomialis*. These species have been implicated in a substantial portion of cases of bacteremia due to *Ainetobacter* spp.—29% in the USA, 24–45% in Taiwan, 50% in Korea, and 66% in Norway [2]. Comparative clinical data indicate that infections caused by *A. baumannii* are associated with more severe symptoms and higher mortality compared with those caused by *A. nosocomialis* [3]. Although *A. calcoaceticus* is part of the complex, it is an environmental species commonly found in soil and water with no established pathogenic role in humans.

The notable differences in pathogenic potential and clinical impact among these closely related species make accurate identification essential. However, due to phenotypic and genotypic similarities, distinguishing Acb complex species remains challenging. Conventional identification methods based on biochemical profiles often lack discriminatory power [4]. Widely implemented automated systems, such as API 20NE, VITEK 2, Phoenix, Biolog, MicroScanWalkAway, frequently fail to distinguish among complex members, leading to misidentification rates of up to 25%, commonly defaulting to *A. baumannii* [5,6]. In this context, matrix-assisted laser desorption ionization-time of flight mass spectrometry (MALDI-TOF MS) and molecular genetic techniques provide an alternative route for accurate taxonomic resolution. Accordingly, this review aims to support the selection of suitable diagnostic approaches for use in routine clinical settings and epidemiological studies.

## 2. Literature Search

The literature search was performed using PubMed and Google Scholar, as well as manual screening for articles published between 2005 and 2024. The used keywords were “identification methods”, “*Acinetobacter baumannii* complex”, and the specific names of the individual approaches. Only English-language publications were considered. In addition, key studies published prior to the main search period were consulted in the sections on taxonomy and genetic methods.

## 3. Historical Background of the Taxonomy of Genus Acinetobacter

*Acinetobacter* species are short pleomorphic Gram-negative rods, defined as coccobacteria, strict aerobes, catalase positive, oxidase negative, non-fermenting, and non-motile [7].

The history of the genus *Acinetobacter* began in 1911 when the Dutch microbiologist Beijerinck isolated a microorganism from soil and named it *Micrococcus calcoaceticus* because of its growth in the presence of calcium acetate [8]. Forty years later, Brisou and Prevot [9] proposed the name *Acinetobacter* (from the Greek “akinetos”—immobile) to distinguish it from the motile organisms in the genus *Achromobacter*. In 1968, Baumann et al. [10] published their detailed study on the taxonomic structure of the genus *Acinetobacter*. In 1986, Bouvet and Grimont [11] created the first classification of the genus *Acinetobacter* based on DNA–DNA hybridization (DDH) studies and distinguished 12 DNA groups or genospecies. Newly in the List of Procaryotic Names with Standing in Nomenclature (LPSN), the genus *Acinetobacter* is presented with 113 species [12]. Most of them are nonpathogenic environmental bacteria.

The term *Acinetobacter calcoaceticus*–*Acinetobacter baumannii* complex was introduced in 1991 by Gerner-Smidt et al. [13] for a group of four phenotypically similar species: *A. calcoaceticus* (genomic species 1), *A. baumannii* (genomic species 2), genomic species 3, and genomic species 13 *sensu* Tjernberg & Ursing [14], which were later named *A. pittii* and *A. nosocomialis*, respectively [15]. In 1993, Gerner-Smidt and Tjernberg [16] reported two other genomic species, each of which included strains isolated from clinical materials. DNA–DNA hybridization studies established the proximity of one species to *A. nosocomialis*, while the other turned out to be close to *A. calcoaceticus* and *A. pittii.* Later, these were referred to as the genomic species “close to 13 TU” and the genomic species “between 1 and 3”, respectively [17,18,19].

Currently, the complex unites the following phenotypically and genetically closely related species: *A. calcoaceticus*, *A. baumannii*, *A. nosocomialis* (former genomospecies 13 *sensu* Tjernberg & Ursing), *A. pittii* (former genomospecies 3), *A. sefertii* (former genomospecies “close to TU13”), and *A. dijkshoorniae* (former genomospecies, between 1 and 3”) [15,20,21]. All these species, except for *A. calcoaceticus*, have been associated with both community- and hospital-acquired infections, exhibiting different resistance patterns.

## 4. Phenotypic Versus Molecular Tests, Rapid Versus High-Resolution Tools

The choice of identification method depends on multiple factors—analytical accuracy, reproducibility, degree of standardization, cost, turnaround time, personnel expertise, and applicability to routine diagnostics or research. MALDI-TOF, as a high- throughput phenotypic technique, is well-suited for clinical laboratories with high sample volumes. PCR-based methods such as loop-mediated isothermal amplification (LAMP) and amplification of the gene encoding OXA-51 β-lactamase (*bla*_OXA-51_) offer rapid, cost-effective solutions, particularly valuable for lower-resource settings [22,23]. Other targeted PCR methods serve as a preliminary step in research [24]. Labor-intensive molecular methods like amplified fragment length polymorphism (AFLP), amplified ribosomal DNA restriction analysis (ARDRA), and sequencing of the β-subunit of the RNA polymerase gene *(rpoB*) were employed to validate MALDI-TOF MS results [25,26]. Although these techniques are now infrequently used, mainly in research, their value as reference methods remains noteworthy. In contrast, whole-genome sequencing (WGS), a high-resolution technique, combined with advances in technology and bioinformatics, is increasingly utilized not only in phylogenetic and epidemiological analyses but also for accurate species identification, as well as resistome and virulome profiling.

## 5. Current Phenotypic Methods for Species Identification in the *Acinetobacter calcoaceticus*–*Acinetobacter baumannii* Complex—Matrix-Assisted Laser Desorption Ionization-Time of Flight Mass Spectrometry (MALDI-TOF MS)

MALDI-TOF MS is increasingly used for the identification of a wide array of microorganisms isolated from clinical specimens [27]. The first study of MALDI-TOF MS as a method for bacterial identification dates from 1975 [28], but it began to be routinely applied after 2000. In 2009, Seng et al. [29] conducted the first identification of clinical isolates by comparing conventional methods and MALDI-TOF MS on three aspects: delay, cost, and level of training of the staff. The researchers concluded that the system was efficient and cost-effective for the rapid identification of microbial isolates.

In the MALDI-TOF MS technique, the sample treated with an energy-absorbing matrix is exposed to a laser beam. This causes ionization of the microbial proteins. Ions are directed to a flight tube and move toward a detector with different velocities according to their mass. The mass spectra generated are characterized by their mass/charge ratio (*m/z*). Unknown organisms are identified by matching the spectra obtained to spectra in the database of the apparatus. Cell extracts as well as bacterial colonies have been tested [30,31]. According to Sousa et al. [26], the use of intact cell samples could achieve better identification for closely related species than cell extracts due to the more complete protein profiles obtained. In addition, the use of intact cell samples can considerably reduce both analysis time and cost [30]. The instrumental analysis itself takes approximately five minutes; thus, the results can be available within 12–24 h after receiving the sample. The system is considered useful for differentiating closely related species that are otherwise indistinguishable by conventional phenotypic methods, offering an inexpensive alternative to laborious and time-consuming molecular techniques [31]. Beyond routine identification, MALDI-TOF MS has also been explored as a typing tool using the open-source MATLAB-based software MicrobesMS 2 [32].

Regarding the ability to distinguish species within the Acb complex, MALDI-TOF MS appears to reliably identify *A. baumannii* and *A. pittii*, but it often misidentifies *A. nosocomialis* and *A. calcoaceticus* [25,33]. To improve the precision of distinguishing genetically close species within the complex, efforts have been made in several directions (Table 1). One such approach is expanding the manufacturers’ reference databases. Espinal et al. [25] found that *A. nosocomialis* strains in their study were erroneously identified as *A. baumannii* due to the absence of a reference spectrum for *A. nosocomialis* in the Bruker database. Jeong et al. [34] added the profiles of 63 additional Acinetobacter strains to the default database. Another study optimized the SARAMIS^TM^ database of the VITEK MS system to improve the identification by increasing the number of reference spectra for each species and creating a so-called ‘SuperSpectrum’ with a high level of confidence of 75% [35]. Following the description of novel species in the complex*,* Mari-Almirall et al. [31] utilized mean spectra (MSP) to enhance the differentiation of *A. nosocomialis*, *A. dijkshoorniae*, and *A. seifertii.*

Sedo et al. [36] proposed an alternative solution to the misidentification of species within the Acb complex. They developed a novel sample preparation protocol by altering the matrix composition. The solvent contained ferulic and formic acid instead of the acids used in the standard protocol, thereby enabling detection of proteins in the high-mass range (over 12 kDa). With the aid of a custom database developed for the study, the alternative protocol enabled improved identification of Acb complex strains (90%) compared with the standard protocol (55%). However, this approach has limitations, as it requires manual control of spectral acquisition and relies on a database that is incompatible with commercial databases.

A third strategy for enhancing the identification of species within the Acb complex is the integration of MALDI-TOF with chemometric tools. It was suggested by Sousa et al. [26]. Using an optimized partial least squares discriminant analysis (PLSDA) and hierarchical cluster analysis (HCA), the authors generated a dendrogram with six distinct clusters within the Acb complex. Each cluster comprised strains of a single species, identified by the selected reference method. The method overcame the misidentification of *A. nosocomialis* as *A. baumannii* and distinguished the species *A. seifertii* and *A. dijkshoorniae* [26]. It is also of note that these results were achieved using intact cells. The authors suggested that intact cell samples may lead to improved identification of closely related species. In contrast, species discrimination was poorer when using cell extracts, likely due to the lower protein yield during the extraction and, consequently, fewer detectable peaks [26].

Toh et al. [37] achieved differentiation of *Acinetobacter* spp. by integrating two TOF mass analyzers via unlicensed open-source software, thereby enhancing the analytical performance of MALDI-TOF MS for protein characterization. Despite representing a step back from automation, the study validated the method’s greater ability to discriminate between closely related species. In addition, the researchers found that differentiation between *A. pittii* and *A. calcoaceticus* was not possible using 16S rRNA or *rpoB* sequencing but was successfully achieved using multiplex PCR targeting the DNA gyrase subunit B gene (*gyrB*) and MALDI-TOF MS [37].

To solidify its position as an identification method of choice, MALDI-TOF MS must keep pace with the evolving classification of bacteria. For this reason, commercial system providers regularly update reference databases. Nonetheless, adding new spectra of species-type strains previously absent from the databases can complicate the discrimination of closely related species. Sedo et al. [38] reported that such database expansion led to misidentification of species already present, due to similarities in their mass spectra. Therefore, each database update should be accompanied by re-validation to ensure that the method’s performance remains uncompromised [38].

Regarding the use of MALDI-TOF MS as a typing method, it must be taken into consideration that proteomic analysis is positioned alongside genomic typing methods; however, the genotype does not always correlate with the phenotype. Therefore, it was concluded that the method is suitable for phenotypic identification but not for studies of clonality [32].

Rim et al. [39] found a poor correlation between dendrograms generated by MALDI-TOF MS and those obtained using the reference method Pulsed-field Gel Electrophoresis (PFGE). The authors attributed this discrepancy to the lack of standardization in sample preparation protocols (the choice of culture media, centrifugation speed, and the use of either cell extracts or intact cells) as well as variability in software platforms. However, MALDI-TOF MS appears to hold promise as an epidemiological typing tool [40]. Busby et al. [41] concluded that MALDI-TOF MS combined with suitable bioinformatic tools could be effectively integrated into outbreak management strategies. Compared with conventional methods, MALDI-TOF MS is a rapid and cost-effective technique capable of identifying isolates at the species level with high accuracy, typically within approximately five minutes per sample. In addition, it can detect virulence determinants and resistance mechanisms, including cefalosporinases, carbapenemases, and even markers of colistin resistance [42,43,44].

Despite its advantages, the approach remains out of reach for healthcare centers with limited financial and personnel resources. In such situations, organizational solutions can facilitate access through regional reference laboratories or consortia. Strains can be forwarded to these facilities for final confirmation following initial phenotypic identification and antimicrobial testing performed on-site.

## 6. Current Molecular-Genetic Methods for Species Identification in the *Acinetobacter calcoaceticus*–*Acinetobacter baumannii* Complex

Molecular-genetic methods remain the most reliable approach to complement the limited phenotypic identification of species within the Acb complex. However, these methods are not yet widely accessible in routine diagnostic laboratories and are primarily employed for taxonomic and research purposes. Although some of these techniques are cost-effective and relatively easy to perform, they enable the identification of a limited subset of bacterial species and often require conventional nutritional, physiological, and microscopic methods prior to differentiation.

Molecular-genetic methods (Table 2) useful for species identification of members in the Acb complex can be grouped as follows:DNA–DNA hybridization (DDH).Restriction analysis of DNA (genomic fingerprinting).PCR-based methods targeting specific genes.Methods based on DNA-sequence analysis.

### 6.1. DNA–DNA Hybridization (DDH)

Over the past 50 years, DDH has been used as the gold standard in taxonomic studies. The first DDH analysis of *Moraxella* and *Moraxella*-like strains assigned them to five DNA groups, with four strains remaining ungrouped [45]. By 1986, twelve DNA groups had been described [11]. Later, two independent studies identified additional DNA–DNA hybridization groups (13 to 17), including 13 sensu Tjernberg and Ursing [14,62]

DDH is considered resource-intensive primarily due to its high demand for technical expertise and time. It is labor-intensive, often needs repetition, and is difficult to standardize [63,64]. Along with the advancement of whole-genome sequencing technologies, efforts have been directed toward correlating traditional DDH values with digital DDH-like similarity indices [65,66]. Of these, average nucleotide identity (ANI) has been the most widely used. ANI was first introduced to mimic the process of experimental DDH and is also referred to as a digital version of DDH [65]. It represents the mean identity of homologous genomic regions shared between two genomes and requires whole-genome sequencing of the strains under investigation. ANI values of 95–96% are generally accepted as corresponding to the traditional 70% DDH threshold for species delineation. Chan et al. [46] resolved inconsistencies in the classification of representative strains within the Acb complex. In their study, *A. nosocomialis* NCTC 10,304 was reclassified as *A. baumannii* NCTC 10,304 based on ANI values exceeding 95% when compared with sequenced *A. baumannii* strains. A correction was also made for *A. calcoaceticus* PHEA-2, which was reclassified as *A. pittii* PHEA-2 [46]. ANI was performed to confirm the species status of *A. seifertii* and *A. dijkshoornia,* recently added to the Acb complex [20,21]. The method is highly reproducible, fast, and easily standardized. It requires access to specialized software tools and a moderate to advanced level of bioinformatic skills, which, at present, may not be readily available in many laboratories. The primary application of ANI remains in research, particularly for confirming novel species or revising existing classification [67,68].

### 6.2. Restriction Analysis of DNA (Genomic Fingerprinting)

The amplified fragment length polymorphism (AFLP) method, first described by Vos et al. [69], involves PCR amplification of restriction fragments generated from digested genomic DNA following adaptor ligation. The resulting fragments are separated by electrophoresis and subsequently analyzed. Janssen et al. [47] demonstrated that AFLP could resolve the taxonomic structure of the genus Acinetobacter and differentiate strains within a species. The method was later adopted as a reference technique for identifying the major *A. baumannii* international outbreak clones I-III [70].

Amplified ribosomal DNA restriction analysis (ARDRA) involves the digestion of the PCR-amplified 16S rRNA gene with restriction enzymes, followed by the analysis of the resulting fragment patterns via gel electrophoresis. This method can identify most *Acinetobacter* species, including members of the Acb complex [48]. Dijkshoorn et al. [49] successfully differentiated *A. baumannii*, *A pittii,* and *A. nosocomialis* using ARDRA with only three restriction enzymes. Furthermore, strains of *A. dijkshoornii* and *A. seifertii* exhibited distinct ARDRA profiles [49]. However, the presence of multiple ARDRA profiles among strains of the same species suggests considerable intraspecies diversity. In a recent study, two commercial MALDI-TOF systems were evaluated for species-level identification within the Acb complex with ARDRA employed as a confirmatory assay [71]. The authors emphasized ARDRA’s validity as a robust identification method and noted its potential to become more cost-effective in the future.

Ribotyping detects genetic polymorphisms within the genes encoding ribosomal RNAs (16S and 23S) and their adjacent DNA sequences [50]. The method consists of genomic DNA digestion with restriction enzymes, followed by electrophoretic separation of the fragments, which are then transferred to a membrane. After hybridization with specific labeled probes, specific species banding patterns are generated. Although labor-intensive, ribotyping is highly discriminatory and reproducible for the identification of *A. calcoaceticus*, *A. baumannii*, *A. pittii*, and *A. nosocomialis,* showing good concordance with AFLP and ARDRA [51].

The cost of the genomic fingerprinting approaches ranges from moderate to high for AFLP and ribotyping and is low for ARDRA. These techniques are generally labor-intensive. While AFLP and ribotyping require skilled personnel, ARDRA can be performed with basic molecular biological skills. Reproducibility depends on the reagents and protocols used; ribotyping automated systems provide good reproducibility. Interlaboratory standardization is generally low. The turnaround time is relatively long, from 1 day in ARDRA and automated ribotyping to 3 days in manual ribotyping. These methods are not used in routine diagnostic practice. Historically, they have been key tools for strain typing, phylogenetic analysis, and epidemiological surveillance. They require basic to intermediate bioinformatic tools and appropriate skills for data handling, analysis, and interpretation [72,73,74].

### 6.3. PCR-Based Methods

The one-tube multiplex PCR is a specific, rapid, and cost-effective technique, first proposed by Chen et al. [51]. Two primer pairs are employed: One targeting a highly conserved 425 bp region of the recombinase A (*recA*) gene present in all *Acintobacter* species. The *recA* fragment serves as an internal control. Another pair of primers is used for amplification of a 285 bp fragment from the ITS region specific to *A. baumannii.* After electrophoresis, the presence of two bands indicates *A. baumannii*, and only one band suggests another *Acinetobacter* species. The method was validated by testing 22 reference strains and 138 clinical isolates. The one-tube multiplex PCR showed 100% agreement with ITS sequencing [51].

Building on the previously described multiplex PCR, the same authors developed a second assay [52] targeting internal fragments of the 16S-23S rRNA intergenic spacer region (16S-23S rRNA ITS), *gyrB,* and *recA*. This method enables the identification of *A. baumannii, A. nosocomialis,* and *A. pittii*. The 425 bp fragment of *recA* is amplified in all *Acinetobacter* isolates. Two bands corresponding to *recA* (425 bp) and *gyrB* (294 bp) are observed in both *A. baumannii* and *A. nosocomialis.* The distinction between these two species is based on the presence of a 208 bp ITS fragment in the ITS region of *A. baumannii. A. pittii* is recognized by the amplification of the *recA* fragment and another 150 bp ITS fragment. Chen et al. [52] validated this method using both pure cultures and positive blood culture media. The assay demonstrated 100% accuracy with pure cultures, while testing with positive blood culture media yielded sensitivity of 92.4% and specificity of 98.5% [52].

The gene encoding the β-lactamase OXA-51 (*bla*_OXA-51_) is a reliable genetic marker for the identification of *A. baumannii*. It is an intrinsic gene that encodes carbapenemase with weak enzymatic activity. However, high-level carbapenem resistance has been associated with the insertion of IS*Aba1* upstream of the gene, which enhances its expression. In the study of Turton et al. [22], a large set of well-characterized clinical isolates was screened for the presence of *bla*_OXA-51_ using PCR with specific primers. All isolates that tested positive for *bla*_OXA-51_ were confirmed as *A. baumannii* by ARDRA. In contrast, PCR-negative isolates were identified as other *Acinetobacter* species. The potential limitation of this method is its inability to detect all variants of the *bla*_OXA-51-_like gene. Additionally, a plasmid carrying an IS*Aba1-bla*_OXA-51-_like gene was identified in a clinical isolate of *A. nosocomialis* [75], raising concerns about the gene’s exclusivity to *A. baumannii.* Zander et al. [76] reported atypical *bla*_OXA-51-_like amplification products in three *A. baumannii* isolates from South Korea, South Africa, and Turkey due to disruptions caused by IS*Aba19* and IS*Aba15* insertions. Similarly, Ahmadi et al. [77] observed IS*Aba19*-mediated disruption of the *bla*_OXA-51-_like gene in 14 *A. baumannii* isolates. These findings indicate that PCR detection of the intrinsic *bla*_OXA-51-_like gene alone may be insufficient for accurate identification of *A. baumannii.* Complementary methods such as sequencing of *rpoB* or multiplex PCR targeting *gyrB* are recommended to improve identification accuracy [76,77].

Two novel rapid PCR-based methods for the identification of *A. baumannii* have been proposed by Abhari et al. [53]. The first one is a duplex PCR targeting both the *bla*_OXA-51-_like gene and the gluconolactonase gene. The latter is highly specific to *A. baumannii,* and it is absent in non-*A. baumannii* species. This approach was designed to overcome the limitations associated with using *bla*_OXA-51-_like as a sole identification marker. The second method is a sequence-specific PCR targeting a 1024 bp fragment of the *rpoB* gene, which is characteristic of *A. baumannii* and has been proposed as a rapid alternative to the more time-consuming *rpoB* gene sequencing. Both methods were evaluated in silico using FastPCR software and in vitro with 210 isolates collected from both abiotic and biotic surfaces. Data analyses revealed that the *rpoB-*based PCR had a sensitivity of 100% and a specificity of 91.18%, while the duplex PCR assay targeting *bla*_OXA-51-_like and gluconolactonase genes demonstrated a sensitivity of 91.06% and a specificity of 100%.

Based on the interspecies heterogeneity observed in the *gyrB*, a multiplex PCR assay was developed to differentiate four closely related species: *A. baumannii*, *A. nosocomialis*, *A. pittii*, and *A. calcoaceticus* [6,54]. Initially, three primers (Sp2F, Sp4R, and Sp4F) were employed to distinguish *A. baumannii* and *A. nosocomialis*. Both species produced a 294 bp amplicon, while a second amplicon of 490 bp was specific to *A. baumannii* [54]. In 2010, an additional multiplex PCR assay was introduced to distinguish *A. calcoaceticus* and *A. pittii*, employing two primer pairs: D16-D8 and D14-D19. A combined multiplex PCR was subsequently performed with all the primers [6]. Using this *gyrB-*based approach, Lee et al. [55] achieved a 100% concordance rate in identifying strains within the Acb complex.

The loop-mediated isothermal amplification (LAMP) method employs strand-displacing Bst DNA polymerase along with a set of internal (FIP, BIP) and external (F3, B3) primers that recognize six distinct regions flanking the target DNA sequence. To enhance the assay’s sensitivity, an additional pair of loop primers (LF, LB) may be included. The amplification is performed under isothermal conditions, and it is typically completed in under one hour. Qualitative results can be obtained either by measuring turbidity caused by a byproduct or by detecting fluorescence from DNA-intercalating dyes [78,79]. These assays can also identify carbapenem resistance depending on the target gene sequence. For the identification of *A. baumannii*, LAMP assays have been developed targeting various genetic markers, including the 16S-23S rRNA ITS [56], *bla*_OXA-51_ [80], *adeS* [81], and *bla*_OXA-23_ [82]. To differentiate *A. baumannii*, *A. pittii*, and *A. nosocomialis*, Sharma et al. [23] employed two genetic markers (the 16S-23S rRNA ITS and the *bla*_OXA-23_) and reported 100% specificity and greater sensitivity of LAMP compared with conventional PCR [82,83]. The RealAmp method integrates LAMP with a portable real-time isothermal scanner [84]. Although it offers improved specificity, it requires additional high-cost reagents and specialized monitoring equipment [85]. The main advantages of LAMP are its low cost, ease of interpretation, and rapid detection, making it particularly suitable for the diagnosis and on-site screening of carbapenem-resistant *A. baumannii* in healthcare settings [23].

PCR-based methods are generally cost-effective, reproducible, and easy to perform. Their turnaround time is within a few hours. These features make them well-suited for routine laboratory applications. While some methods, such as *bla*_OXA-51_ PCR, are fully standardized and routinely used, others, like *recA-*based PCR techniques, exhibit only moderate standardization. This refers to the existence of protocols in certain studies but a lack of widespread interlaboratory validation. These assays typically do not require bioinformatic tools except in cases where custom primer design is necessary. In such cases, tools like Primer-BLAST are largely accessible to laboratories with limited recourses [51,86,87].

### 6.4. Methods Based on the Analysis of DNA Sequences

DNA sequences of genes encoding 16S rRNA, *rpoB*, and the 16S-23S ITS have been widely used for the classification and identification of species within the Acb complex.

The 16S rRNA gene is commonly used in bacterial taxonomy due to its highly conserved structure. Sequencing results are typically compared with those of known type strains in public databases such as GenBank. Misbah et al. [88] amplified a 1050 bp region to identify clinical isolates of the genus *Acinetobacter*. Although this technique is useful for bacterial identification, it is not effective in differentiating *Acinetobacter* species because of the extremely low polymorphism in the variable regions of the 16S rRNA gene. Therefore, alternative genetic markers such as *rpoB*, *gyrB*, and *recA* genes are recommended for improved resolution [57]. The intraspecies similarity of the 16S rRNA gene within the Acb complex has been reported to exceed 99.8%, while interspecies similarity ranges from 97 to 98% [20]. To improve the discriminatory power of this method, a revised threshold of 99.7% interspecies similarity has been proposed for species-level identification [89]. In a comparative study by Lee et al. [55], 16S rRNA gene sequencing misidentified seven isolates of *A. calcoaceticus* as *A. nosocomialis* or *A. pittii*, three of nine *A. pittii* isolates as *A. nosocomialis*, and one *A. baumannii* isolate as *A. nosocomialis*. Overall, these findings indicate low sensitivity for *A. calcoaceticus* and low specificity for *A. nosocomialis*.

The *rpoB* gene has two polymorphic regions—region 1 (350 bp) and region 2 (450 bp) flanked by two variable intergenic spacers: *rpl*L*-rpoB* (301–310 bp) and *rpoB-rpo*C (86–177 bp), which vary among species. By designing primers to amplify these four fragments, La Scola et al. [57] developed a method termed partial *rpoB* sequencing (PRBS) for the identification of *Acinetobacter* species. Gundi et al. [58] validated a PRBS protocol based on PCR amplification and sequencing of polymorphic region 1 using a large collection of previously well-characterized strains. Their study demonstrated clear separation of species within the Acb complex, with interspecies similarity varying from 88.3% to 96.9% and intraspecies similarity from 98% to 100% [58]. Subsequent studies confirmed the utility of PRBS for *Acinetobacter* species identification [55,90]. To date, PRBS remains one of the most effective methods for differentiating species within the genus *Acinetobacter* [91]. Moreover, most studies evaluating and validating MALDI-TOF MS for species-level identification within the Acb complex used *rpoB* sequencing as a confirmatory reference method [26,31,35,36].

The 16S-23S rRNA intergenic spacer (ITS) region is a genetically unstable locus with variable length. While the ITS size shows minimal variation among strains within the same species, significant differences can be observed between species of the same genus [91]. Using ITS sequencing, Chang et al. [59] evaluated 11 reference strains from the Acb complex and 17 additional reference strains belonging to the genus *Acinetobacter*. They observed sequence similarity within species ranging from 99% to 100% and between species ranging from 86% to 92%. During the validation phase, 76 out of 79 (96.2%) clinical isolates of the Acb complex were correctly identified, demonstrating the method’s strong discriminatory power [59].

PRBS and ITS sequencing are primarily utilized in reference settings due to their moderate to high cost, a turnaround time of 1–2 days, and the requirement for basic informatics tools for sequence analysis—factors that may limit their routine diagnostic application. PRBS provides superior resolution and has broader validation, whereas ITS-sequencing is less standardized and shows greater variability in performance. Considering these aspects, PRBS is generally more suitable when high-resolution identification is required beyond routine diagnostic workflows [92,93].

The multilocus sequence analysis (MLSA) is a genomic approach employed to determine clonal relationships among bacterial strains. It involves sequencing internal fragments of seven housekeeping genes—*cpn60*, *fusA*, *gltA*, *pyrG*, *recA*, *rplB*, and *rpoB*—which are also commonly used in multilocus sequence typing (MLST) [94]. To delineate the phylogenetic positions of *A. nosocomialis* and *A. pittii*, Nemec et al. [15] characterized 80 strains representative of the diversity within the Acb complex. This study incorporated MLSA in conjunction with other molecular methods. The results revealed distinct clusters corresponding to each species, with intra- and interspecies sequence similarities supporting their genetic distinctiveness. However, some individual genes failed to resolve species boundaries accurately in phylogenetic trees. Consequently, reliance solely on MSLA may lead to erroneous taxonomic assignments; for instance, *A. calcoaceticus* and *A. dijkshoorniae* were incorrectly grouped as a single species in certain analyses [15].

MLSA involves moderate to high cost and a turnaround time of 1–2 days, provided sequence data are readily available. The method requires intermediate level bioinformatic tools for sequence alignment and phylogenetic analysis. MLSA is considered reproducible and relies on established protocols; however, standardization between laboratories remains limited due to variability in gene targets and thresholds for species delineation. MLSA is primarily applied in reference laboratories and research settings for epidemiological investigations and taxonomic studies [95].

Whole-genome sequencing (WGS) refers to the process of determining the complete DNA sequence of an organism’s genome [60]. This technology has changed the practice of clinical bacteriology and public health laboratories by providing comprehensive insights into the genomic characteristics of bacterial pathogens. The Sanger method, also known as the chain termination method, is considered the reference technique for DNA sequencing [61]. Since 2004, the development of next-generation sequencing (NGS) technologies has enabled the simultaneous sequencing of millions of DNA fragments in a single run, offering up to 100-fold increase in speed compared with the Sanger approach [96]. The implementation of NGS platforms has revolutionized bacterial genome research through their high throughput, reduced cost, and improved time efficiency [97].

NGS methods are broadly categorized into short-read (read length of 50–500 bp) and long-read (read length of 10 to >50 kb) technologies. Pyrosequencing is a short-read method that determines DNA sequences by detecting the release of pyrophosphate during nucleotide incorporation into the growing DNA strand. A widely used short-read platform developed by Illumina/Solexa utilizes sequencing-by-synthesis (SBS), which relies on reversible dye terminators for base detection [98]. Despite their widespread use, short-read sequencing methods present several limitations, including the following:Challenges in de novo assembly due to the limited length of read fragments (≤500 bp);Difficulty in accurately sequencing genomic regions with high or low GC content, tandem repeats, or interspersed repetitive elements;Incomplete de novo assemblies potentially omitting essential genomic regions or genes as a result of DNA fragmentation during library preparation [99].

Oxford Nanopore technology is a long-read sequencing method in which single-stranded DNA molecules are passed through nanopores and changes in electrical current are measured to determine the nucleotide sequence [100]. The long reads produced by this approach enable effective resolution of repetitive regions within bacterial genomes and facilitate the generation of high-quality complete genome assemblies, including plasmids as distinct nucleotide sequences. However, a notable limitation of this technology is its higher error rate compared with SBS methods [101].

Analysis of sequencing data is carried out using a range of bioinformatic tools, which continue to evolve alongside advances in sequencing technologies. In a review of Mustafa [61], the analytical workflow is systematized into the following steps:Data cleaning: quality assessment, trimming of low-quality reads, removal of contaminants;De novo assembly: construction of contiguous sequences from raw reads;Scaffolding: ordering and orienting contigs using paired-end information;Assembly quality assessment: evaluation of metrics such as N50, genome completeness, and contamination;Annotation: identification of functional genomic elements along the sequence of the genome;Read mapping: alignment of sequence reads to a reference genome;Variant calling: detection of single-nucleotide polymorphisms (SNPs), insertions, and deletions;Core genome analysis: identification of conserved genes shared among strains;Strain typing: determination of sequence types often using MLSTAntimicrobial resistance predictionPhylogenetic analysisTree visualization

The strains *A. baumannii* ATCC17978 and *A. baumannii* AYE were the first representatives of the genus *Acinetobacter* to undergo WGS using pyrosequencing and the Sanger method, respectively [102,103]. The sequencing results revealed the complete genomic structures, including genomic islands harboring resistance genes to antimicrobials and heavy metals, as well as resistance genes located outside these islands. Genes encoding secretion systems, efflux pumps, and numerous mobile genetic elements flanking resistance determinants were also identified. Moreover, WGS enabled the identification of horizontally acquired DNA from other pathogenic bacteria and facilitated comparative analyses with nonpathogenic *A. baumannii* strains. These studies highlighted the organism‘s capability for horizontal gene transfer and underscored its role in pathogenicity and antimicrobial resistance [102,103,104]. Later, other epidemic *A. baumannii* strains were subjected to WGS, further enhancing our understanding of the genomic organization of the pathogen [105,106].

Currently, WGS serves as a powerful tool for resistome profiling and assessing clonal relatedness among *A. baumannii* isolates [24,107,108]. It is widely applied in epidemiological investigations at both global and hospital outbreak levels. The single-nucleotide polymorphisms (SNPs) typing and the core genome multilocus sequence typing (cgMLST), performed following WGS, offer significantly higher resolution compared with conventional MLST approaches [109,110]. The continuous refinement of the WGS technology has established it as a preferred method not only for research but also in clinical microbiology for isolate identification, antimicrobial resistance (AMR) profiling, and outbreak analysis [111]. It has already been successfully used to identify bacterial isolates from pure cultures [112] and directly from clinical specimens [113,114,115]. In support of these applications, web-based platforms have been developed for the identification and genotyping of pathogenic bacteria [116,117].

WGS, performed by NGS technologies, entails high cost and a turnaround time of 1–5 days, depending on sequencing platforms and pipeline efficiency. It requires advanced bioinformatic tools and specialized expertise for analysis and interpretation. While reproducibility and accessibility have improved, standardization remains limited. WGS is widely used in reference laboratories and epidemiological studies, but its adoption in routine diagnostics remains constrained by infrastructural and financial demand [118,119].

## 7. Conclusions

There are clinically and epidemiologically relevant differences among the species within the Acb complex, making accurate species-level differentiation essential.

Beyond guiding method selection, the review addresses challenges related to the availability and implementation of the discussed techniques. Although MALDI-TOF MS is increasingly displacing PCR-based assays in routine laboratory settings, the diagnostic potential of some of these traditional methods may still be realized through advances in technological platforms and software tools. Another promising direction involves exploring the potential of MALDI-TOF MS as a tool for epidemiological surveillance. Future research may focus on integrating WGS as a real-time method in healthcare settings, enabling its application in the routine diagnostic workflows and supporting hospital infection control and management.

## Figures and Tables

**Table 1 microorganisms-13-01819-t001:** Comparative assessment and improvement of MALDI-TOF MS for identification of species in the ABC complex.

Accuracy Rate with Default Databases	Accuracy Rate After Improvement	Comparative Methods Used	Number of Validation Strains	Improvement (notes)	References
70% correct ID up to species level	98% correct ID up to species level	ARDRA, ITS, recA, and *bla*_oxa-51_ analysis by sequencing or PCR and gel electrophoresis	*A. pittii*—17 *A. nosocoialis*—18 *A. baumannii*—18	Expanding the database with reference spectra. (Before inclusion of reference spectra for *A. nosocomialis*, strains were misidentified as *A. baumannii*).	Espinal et al., 2012 [25].
69.8% correct ID up to species level	100% correct ID up to species level	*rpoB* and 16SrRNA sequencing, *bla*_oxa-51_PCR analysis	*A. baumannii*—419 *A. nosocomialis*—36 *A. pittii*—23	Expanding the database with reference spectra. (Before inclusion of reference spectra, 49/110 *A. nosocomialis* and 5/78 *A. pittii* were misidentified as *A. baumannii*).	Jeong et al., 2016 [34].
PPV of 56.5%	PPV of 99.6% by use of cell extracts. PPV of 96.8% by use of colonies	*rpoB* sequencing, MLSA	*A. baumannii*—16 *A. nosocomialis*—24 *A. pittii*—15 *A. dijkshoorniae*—12 *A. seifertii*—11	Expanding the database with reference spectra and MSP (Before inclusion of MSP 13/24 *A. nosocomialis* were misidentified as *A. baumannii*; 12/12 *A. dijkshoorniae* were misidentified as *A. pittii*, and 11/11 *A. seifertii* were misidentified as *A. baumannii*).	Mari-Almirall et al., 2017 [31].
Not reported	100% consistency with *rpoB* sequencing	*rpoB* sequencing, MLSA	100 Acb complex isolates	Expanding the database with reference spectra and SuperSpectra.	Pailhories et al., 2015 [35].
86% correct ID up to species level	90% correct ID up to species level	*rpoB* and 16SrRNA sequencing*,* MLSA, ribotyping	*A. baumannii*—32 *A. nosocomialis*—29 *A. pittii*—22 *A. calcoaceticus*—22	Replacement of CHCA and TFA with FerA and FA in the matrix solution.	Sedo et al., 2013 [36].
Sensitivity/specificity (%)		*rpoB* sequencing, MLSA AFLP		Chemometric analysis of MALDI-TOF data	Sousa et al., 2014 [26]
*A. baumannii* 100/89	*A. baumannii* 100/100		*A. baumannii*—28		
*A. nosocomias* 90/100	*A. nosocomialis* 100/95		*A. nosocomialis*—20		
*A. pittii* 100/98	*A. pittii* 100/100		*A. pittii*—20		
*A. calcoaceticus* 100/99	*A. calcoaceticus* 89/100		*A. calcoaceticus*—9		
*A. dijkshoorniae* N.A	*A. dijkshoorniae* 100/100		*A. dijkshoorniae*—4		
*A. seifertii* N.A.	*A. seifertii* 100/100		*A. seifertii*—2		

ID—Identification; PPV—positive predictive value; MSP—mean spectra; N.A.—not applicable; ARDRA—Amplified ribosomal DNA restriction analysis; MLSA—multilocus sequence analysis; ITS—intergenic spacer region. CHCA—Alpha-cyano-4-hydroxycinnamic acid; TFA—Trifluoroacetic acid; FerA—Ferulic acid; FA—Formic acid.

**Table 2 microorganisms-13-01819-t002:** Summary of molecular methods for identification of species within the ABC complex.

The Principle of Method	Method	Identified Species	References
DNA–DNA hybridization	DNA–DNA hybridization	*A. calcoaceticus*, *A. baumannii* *A. nosocomialis* *A. pittii* *A. seifertii* *A. dijkshoorniae*	Johnson et al., 1970 [45]; Bouvet & Grimont, 1986 [11]; Tjernberg & Ursing, 1989 [14]; Nemec et al., 2011, 2015 [15,20]; Cosgaya et al., 2016 [21].
	Average Nucleotide Identity (ANI)	*A. calcoaceticus* *A. baumannii* *A. nosocomialis* *A. pittii* *A. seifertii* *A. dijkshoorniae*	Chan et al., 2012 [46]; Nemec et al., 2011, 2015 [15,20]; Cosgaya et al., 2016 [21].
Restriction analysis of DNA (genomic fingerprinting)	Amplified fragment length polymorphism (AFLP)	*A. calcoaceticus* *A. baumannii* *A. nosocomialis* *A. pittii*	Janssen et al., 1997 [47]; Nemec et al., 2011 [15].
	Amplified ribosomal DNA restriction analysis (ARDRA)	*A. calcoaceticus* *A. baumannii* *A. nosocomialis* *A. pittii* *A. seifertii* *A. dijkshoorniae*	Vaneechoutte et al., 1995 [48]; Dijkshoorn et al., 1998 [49].
	Ribotyping	*A. calcoaceticus* *A. baumannii* *A. nosocomialis* *A. pittii*	Gerner-Smidt, 1992 [50]; Chen et al., 2007 [51].
PCR-based methods	Amplification of a 425-bp fragment of *recA* and a 285-bp fragment of ITS	*A. baumannii*	Chen et al., 2007 [51].
	Amplification of a 425-bp fragment of *recA*, 208-bp and 150bp fragments of ITS, and 294-bp fragment of *gyrB*	*A. baumannii* *A. nosocomialis* *A. pittii*	Chen et al., 2014 [52].
	Amplification of *bla*_OXA-51-_like gene	*A. baumannii*	Turton et al., 2006 [22].
	Amplification of *bla*_OXA-51-_like and gluconolactonase genes	*A. baumannii*	Abhari et al., 2021 [53].
	Amplification of a 1024-bp region of *rpoB*	*A. baumannii*	Abhari et al., 2021 [53].
	*gyrB* gene multiplex PCR	*A. calcoaceticus* *A. baumannii* *A. nosocomialis* *A. pittii*	Higgins et al., 2007, 2010 [6,54]; Lee et al., 2014 [55].
	Loop-mediated isothermal method (LAMP)	*A. baumannii* *A. nosocomialis* *A. pittii*	Soo et al., 2013 [56]; Sharma and Gaind, 2021 [23].
Methods based on the analysis of DNA sequences	16S rRNA gene sequencing	*A. baumannii* *A. nosocomialis* *A. pittii*	La Scola et al., 2006 [57]; Nemec et al., 2015 [20]; Lee et al., 2014 [55].
	Partial *rpoB* sequencing (PRBS)	*A. calcoaceticus* *A. baumannii* *A. nosocomialis* *A. pittii* *A. seifertii* *A. dijkshoorniae*	Gundi et al., 2009 [58]; Nemec et al., 2015 [20]; Cosgaya et al., 2016 [21].
	16S-23S intergenic spacer (ITS) sequencing	*A. calcoaceticus* *A. baumannii* *A. nosocomialis* *A. pittii*	Chang et al., 2005 [59].
	Multilocus sequence analysis (MLSA)	*A. baumannii* *A. nosocomialis* *A. pittii* *A. seifertii* *A. calcoaceticus* *A. dijkshoorniae*	Nemec et al., 2011 [15].
	Whole-genome sequencing (WGS)	*A. calcoaceticus* *A. baumannii* *A. nosocomialis* *A. pittii* *A. seifertii* *A. dijkshoorniae*	Didelot et al., 2012 [60]; Mustafa, 2024 [61].

## Data Availability

No new data were created or analyzed in this study. Data sharing is not applicable to this article.

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
