# Peer review of "Current Methods for Reliable Identification of Species in the Acinetobacter calcoaceticusAcinetobacter baumannii Complex"

_microorganisms, 2025, doi:10.3390/microorganisms13081819_

Round 1

Reviewer 1 Report

Comments and Suggestions for Authors

Comments and suggestions

  1. The paper states that “this review … summarizes current phenotypic and genotypic methods”  yet offers no guiding questions, inclusion rationale or viewpoint (diagnostic, epidemiologic, laboratory). Specify clear objectives and a conceptual frame to steer the narrative.
  2. Narrative reviews are not required to follow PRISMA, but citing core databases, time span and key terms is considered good practice. A concise “Literature Search” paragraph would enhance transparency.
  3. Classic studies from the 1970s appear alongside 2023-2025 papers, but no cut-off date is given. Stating, for instance, “literature up to March 2025” helps readers gauge currency.
  4. The lengthy taxonomic background on the Acb complex could be condensed, freeing space for a critical comparison of present-day methods and their laboratory feasibility.
  5. Table 2 groups highly diverse techniques (DDH, ANI, PCR, WGS) without weighing their relative strengths (turnaround, cost, bioinformatics needs). Add comparative commentary to guide method selection.
  6. Fast and low-cost features of MALDI-TOF are noted, yet implementation challenges in low- and middle-income laboratories are not addressed. Suggest practical strategies (reference hubs, regional consortia).
  7. Some performance figures seem taken from earlier reviews rather than original studies. Cross-verify numerical data with primary sources to avoid error propagation.
  8. Narrative reviews are vulnerable to subjective study choice. State this limitation and explain mitigation efforts (e.g., hand-searching recent reference lists).
  9. Creating sub-sections (phenotypic vs molecular; rapid tests vs high-resolution tools) would streamline the text and prevent overlap between sections 3 and 4.
  10. The manuscript notes database-related misidentification issues; include quantitative examples and recommended re-validation steps drawn from manufacturer guidance or literature.
  11. The conclusion should highlight current gaps (e.g., limited real-time WGS availability) and propose future research avenues, not merely restate previous sections.
  12. While conflicts are declared absent, the review’s funding status is not mentioned. Disclosing any support (or affirming none) bolsters trustworthiness.

Author Response

Dear reviewer 1,

Thank you for your constructive review.

Comment 1: The paper states that “this review…..summarizes current phenotypic and genotypic methods” yet offers no guiding questions, inclusion rationale or viewpoint (diagnostic, epidemiologic, laboratory). Specify clear objectives and a conceptual frame to steer the narrative.

Response 1: Thank you for pointing this out. We revised the text to clarify the objectives and provide a clearer conceptual frame. We applied these corrections to the end of the abstract and introduction.

Comment 2: Narrative reviews are not required to follow PRISMA, but citing core databases, time span and key terms is considered good practice. A concise “Literature Search” paragraph would enhance transparency.

Response 2: We appreciate the note. We included a paragraph “Literature search”.

Comment 3: Classic studies from the 1970s appear alongside 2023-2025 papers, but no cut-off date is given. Stating, for instance, “literature up to March 2025” helps readers gauge currency.

Response 3: Thank you. In response, we addressed it in the paragraph “Literature search”.

Comment 4: The lengthy taxonomic background on the Acb complex could be condensed, freeing space for a critical comparison of present-day methods and their laboratory feasibility.

Response 4: We shortened the section Taxonomy based on your suggestion.

Comment 5: Table 2 groups highly diverse techniques (DDH, ANI, PCR, WGS) without weighing their relative strengths (turnaround, cost, bioinformatics needs). Add comparative commentary to guide method selection.

Response 5: Thank you for this comment. In response, we applied an additional text for each method concerning comparative commentary to guide method selection.

Comment 6: Fast and low-cost features of MALDI-TOF are noted, yet implementation challenges in low- and middle-income laboratories are not addressed. Suggest practical strategies (reference hubs, regional consortia).

Response 6: We appreciate this recommendation. We outlined practical strategies for low- and middle-income laboratories in the last paragraph of Section 5.

Comment 7: Some performance figures seem taken from earlier reviews rather than original studies. Cross-verify numerical data with primary sources to avoid error propagation.

Response 7: We have revised ones again all original studies and verified their findings. Also, we clarified the sentence on a line 31.

Comment 8: Narrative reviews are vulnerable to subjective study choice. State this limitation and explain mitigation efforts (e.g., hand-searching recent reference lists).

Response 8: You are right. We noted hand-searching in the section “Literature search.”

Comment 9: Creating sub-sections (phenotypic vs molecular; rapid tests vs high-resolution tools) would streamline the text and prevent overlap between sections 3 and 4.

Response 9: In accordance to your note, we provided a comparative commentary on the methods in this context, placing it as a transitional section between the taxonomy overview and the narrative on MALDI-TOF MS.

Comment 10: The manuscript notes database-related misidentification issues; include quantitative examples and recommended re-validation steps drawn from manufacturer guidance or literature.

Response 10: We deliberately limited the inclusion of validation data to maintain a more concise and focused narrative.

Comment 11: The conclusion should highlight current gaps (e.g., limited real-time WGS availability) and propose future research avenues, not merely restate previous sections.

Response 11: We agree. We revised the conclusion to outline directions for future research and practical implementation.

Comment 12: While conflicts are declared absent, the review’s funding status is not mentioned. Disclosing any support (or affirming none) bolsters trustworthiness.

Response 12: Yes, we disclosed the funding provided by Medical University of Pleven.

Reviewer 2 Report

Comments and Suggestions for Authors

The manuscript "Current Methods for Reliable Identification of Species in the Acinetobacter calcoaceticus-Acinetobacter baumannii Complex" by Marinova-Balgaranova et al. represents a valuable contribution to the field of medical microbiology. Acinetobacter spp. is a more and more frequently found pathogen in current practice and correct identification to the level of species is sometimes challenging, even in bigger laboratories with appropriate equipment. The authors do a great job in summarizing the recent literature regarding this aspect.

The manuscript is well-written. It has a logical flow, the subsections are divided clearly and the text is written in an easy to understand manner. The tables are a great addition to the work as they properly summarize the information and make it easily accessible to the reader. 

I believe the manuscript fills a gap in literature and raises some concerns regarding accurate species identification using MALDI-TOF and molecular methods and does a good job detailing both of them and presenting the advantages and disadvantages of each of them.

However, I have some suggestions regarding English language which I have detailed below.

Comments on the Quality of English Language

The quality of the English language needs to be improved. There are several places where the sentences have a very strange word order especially in the first section (lines 23-49). Some examples of correction (but not limited to, so please check further):

  • line 26 - please remove the from "the WHO"
  • line 35 - Although A. calcoaceticus is part of the Acb complex, it is primarily environmental species commonly found in soil and water and has not been linked to serious clinical diseases. - It is a primarily environmental... - please correct
  • Acinetobacter is not always written in italics - e.g., lines 57, 59, 66, 68, 240, 447 etc. Please check everywhere throught the text and correct
  • lines 150-151 - please replace "manuel" with "manual"
  • line 215 - please write Moraxella in italic. Also check throught the text

Author Response

Dear reviewer 2,

We are very grateful for your positive evaluation of our article and truly appreciate your feedback.

We fully agree with your comments and have taken the following actions:

  1. We corrected the English in the indicated sections and additionally rechecked the manuscript for other inappropriate expressions.
  2. Thank you for noting the formatting of the genus name. We italicized it through the text as required.
  3. We removed “the” from WHO – line 26
  4. According to your remark, we corrected the sentence ……..– line 35
  5. We replaced “manuel” with “manual” – lines 150-151.

We are very sorry. All these are our mistakes.

Thank you again!

Round 2

Reviewer 1 Report

Comments and Suggestions for Authors

The authors have responded to my comments and suggestions to the best of their ability. I have no further comments on this revision.